# Prognostic Value of the Serum HER2 Extracellular Domain Level in Breast Cancer: A Systematic Review and Meta-Analysis

**DOI:** 10.3390/cancers14194551

**Published:** 2022-09-20

**Authors:** Yun Wu, Lixi Li, Di Zhang, Fei Ma

**Affiliations:** Department of Medical Oncology, National Cancer Center/National Clinical Research Center for Cancer/Cancer Hospital, Chinese Academy of Medical Sciences and Peking Union Medical College, No. 17, Panjiayuan Nanli, Chaoyang District, Beijing 100021, China

**Keywords:** breast cancer, HER2 ECD, prognosis, meta-analysis

## Abstract

**Simple Summary:**

We performed a systematic review and meta-analysis evaluating the predictive value of the serum HER2 extracellular domain (sHER2 ECD) for breast cancer prognosis. Our review investigated 40 studies (12,229 patients) that assessed the impact of the sHER2 ECD levels on breast cancer prognosis and explored the clinical significance of sHER2 ECD levels in different treatment modalities. Our findings indicated that an elevated sHER2 was an unfavorable prognostic factor in breast cancer. More interestingly, sHER2 ECD was found to be a promising biomarker for predicting adverse clinical outcomes of trastuzumab-based treatment but did not affect the efficacy of tyrosine kinase inhibitors. In addition, the baseline cutoff value of sHER2 ECD in different treatment stages of breast cancer remains to be further explored. Overall, our study suggested that sHER2 ECD levels have important prognostic value in breast cancer and may be helpful for clinicians to select the appropriate anti-HER2 therapy for HER2-positive breast cancer, providing more evidence for guiding clinical practice.

**Abstract:**

An elevated serum HER2 extracellular domain is associated with poor prognosis in breast cancer, but the relationship between sHER2 and the efficacy of different modalities remains controversial. Herein, we aimed to evaluate the prognostic value of serum HER2 extracellular domain (sHER2 ECD) in breast cancer and to identify its correlation with the efficacy of different treatment regimens. A systematic search of the PubMed, Embase, Cochrane Library, Web of Science, and Scopus databases was conducted to identify studies exploring the association between HER2 ECD level and clinical outcomes among patients with breast cancer. Using the random effects models, pooled hazard ratios (HRs), and odds ratios (ORs) with 95% confidence intervals (CI), were calculated for progression-free survival (PFS), overall survival (OS), disease-free survival (DFS), and the objective response rate (ORR). Heterogeneity was further evaluated by subgroup and sensitivity analysis. Overall, 40 studies comprising 12,229 patients were included in this systematic review and meta-analysis. Elevated HER2 ECD levels were associated with worse PFS (HR 1.74, 95% CI 1.40–2.17; *p* < 0.001), and this effect was observed in patients treated with chemotherapy (HR 1.81, 95% CI 1.37–2.39; *p* < 0.001), endocrine therapy (HR 1.91, 95% CI 1.57–2.32; *p* < 0.001), and trastuzumab (HR 1.74, 95% CI 1.31–2.30; *p* < 0.001). However, this association was not present in patients treated with tyrosine kinase inhibitors (TKIs) (HR 1.44, 95% CI 0.85–2.43, *p* = 0.17). The HRs/ORs for an elevated HER2 ECD level for DFS, OS, and ORR were 2.73 (95% CI 2.17–3.42; *p* < 0.001), 2.13 (95% CI 1.77–2.57; *p* < 0.001), and 0.80 (95% CI 0.49–1.31; *p* = 0.381), respectively. An elevated sHER2 ECD was an unfavorable prognostic factor in breast cancer but did not affect the efficacy of tyrosine kinase inhibitors such as lapatinib. Detection of sHER2 ECD may be helpful for clinicians selecting the appropriate anti-HER2 therapy for patients with HER2-positive breast cancer.

## 1. Introduction

Breast cancer is the most common type of cancer in women worldwide and the leading cause of cancer-related death in females [1]. In 2022, roughly 287,850 new cases of female breast cancer will be diagnosed, and there will be an estimated 43,250 deaths from female breast cancer [2]. Of these, approximately 14% of all cases are classified as the human epidermal growth factor receptor 2 (HER2) positive subtype [2], which is generally associated with a more aggressive and worse prognosis [3]. HER2 is a tyrosine kinase receptor protein with a molecular weight of 185 kDa expressed by a proto-oncogene. Its protein contains three domains, namely the extracellular domain (ECD), the intracellular segment with tyrosine kinase activity, and the transmembrane region [4]. Patients with HER2 overexpression are usually less sensitive to chemotherapy agents; however, the application of therapeutic strategies targeting HER2 has introduced more treatment options for these patients [5,6,7]. Current anti-HER2 therapy includes a variety of therapeutic regimens, such as monoclonal antibodies, small molecule tyrosine kinase inhibitors (TKIs), and antibody-drug conjugates. However, drug resistance is inevitable during treatment, and exploring biomarkers that can predict the efficacy of different anti-HER2 treatments is an important way to improve the efficacy.

HER2 ECD has been shown to be released into the blood from cancer cells and can be measured in serum [4]. Due to the limitations of subjectivity and hysteresis in the detection of tissue HER2 expression [8] and the fact that serum HER2 ECD (sHER2 ECD) has the advantage of convenience and accessibility, more studies have focused on sHER2 ECD [9,10]. An average of 18.5% of patients with primary breast cancer have elevated sHER2 ECD, compared with approximately 23–80% of patients with metastatic disease [11]. Notably, sHER2 ECD plays multiple roles in assisting diagnosis, facilitating real-time assessment, and evaluating prognosis [11,12]. sHER2 ECD serves as a highly sensitive and specific biomarker for breast cancer screening [13]. More importantly, it has also been shown to be an independent prognostic factor in HER2-positive breast cancer [14]. Different cutoff values have been proposed [15], and sHER2-ECD >15 ng/mL is often used as the positive evaluation standard [14]. An elevated sHER2 ECD has been shown to be associated with poor progression-free survival (PFS), and it was also an unfavorable predictor for anti-HER2 therapy [16]. In addition, Darlix et al. also found that metastatic breast cancer patients with increased sHER2 ECD had a worse prognosis [17]. Despite these promising prospects, the practical application of sHER2 ECD in the management of breast cancer has been hampered by the lack of inconsistent results [18]. Differences in enrolment populations, study designs, detection methodologies, tumor heterogeneity, and cutoff values among different studies may complicate the assessment of its prognostic value [19,20]. It is necessary to synthesize the available evidence from these studies to explore the potential effects of sHER2 ECD levels in patients with HER2-positive breast cancer.

Thus, a systematic review and meta-analysis were performed to evaluate the prognostic impact of baseline sHER2 ECD levels and to further investigate the clinical significance of sHER2 ECD in different treatment modalities, providing more evidence for guiding clinical practice.

## 2. Materials and Methods

### 2.1. Systematic Literature Search

This systematic review and meta-analysis were conducted in accordance with the Preferred Reporting Items for Systematic Reviews and Meta-analysis (PRISMA) guidelines [21]. The study protocol was registered on PROSPERO (Registration Number: CRD42022349499). A comprehensive literature search was performed in the PubMed, Embase, Cochrane Library, Web of Science, and Scopus databases before 11 September 2022. Publications with the following search keywords were included: breast neoplasms, HER2, and extracellular domain. The detailed search strategy is elaborated in Appendix A.

### 2.2. Inclusion and Exclusion Criteria

The inclusion criteria of the present analysis were as follows: (1) Population: female with breast cancer regardless of clinical setting. (2) Intervention: anti-tumor treatment including endocrine therapy, anti-HER2 therapy, and adjuvant therapy. (3) Comparison: comparison of prognosis between patients with elevated sHER2 ECD level or non-elevated sHER2 ECD level. (4) Outcomes: the primary outcomes PFS and the secondary outcomes were disease-free survival (DFS), overall survival (OS), and objective response rate (ORR). The hazard ratio (HR) and 95% confidence interval (CI) for the multiple outcomes should be reported. (5) Study design: randomized controlled trial (RCT), cohort study, and case control study will be included. Exclusion criteria for the study were: (1) non-breast cancer patients; (2) no relevant outcomes; (3) reviews/meta-analyses, letters, comments, editorials or case reports; (4) non-human research studies such as animal or experimental studies; (5) non-English articles.

### 2.3. Data Extraction and Quality Assessment

Three authors independently (YW, LL, and DZ) conducted the study selection and data extraction, and any discrepancies were resolved through discussion. Certain information was extracted from the qualifying publications: author names, year of publication, ethnicity, sample size, study design (prospective or retrospective), disease status (neoadjuvant, adjuvant, and metastatic), treatment, cutoff values, exposure assessment, and outcomes (ORR, DFS, PFS, and OS). PFS was the primary outcome. DFS, OS, and ORR were the secondary outcomes. Efforts were made to reach the authors in circumstances when the aforesaid information was absent from the articles. The definitions of endpoints were listed as follows: (1) PFS: the time from treatment to progression or death from any cause; (2) OS: the time from surgery or diagnosis to death from any cause; (3) DFS: the time from surgery or diagnosis to relapse or death from any cause; and (4) ORR: the sum of complete and partial response.

The Newcastle–Ottawa Scale (NOS) was used to evaluate the quality of the nonrandomized studies by two authors (YW and LL) independently [22]. Briefly, this system evaluates studies based on the following three domains: selection of participants (4 items); comparability between groups (1 item); and exposure assessment (3 items). Studies with NOS scores of ≥7 were considered high-quality studies, while those with scores less than 7 were assessed as low-quality studies.

### 2.4. Statistical Analysis

The pooled HR and odds ratios (OR) with 95% CI were calculated for each outcome using the random-effects model. Heterogeneity between the studies was assessed using Cochran’s Q test and I^2^ statistics. I^2^ values of ≥50% are generally considered to indicate substantial heterogeneity. We performed subgroup analysis to evaluate whether the effects of sHER2 ECD differed for the treatment modalities (endocrine therapy, anti-HER2 therapy, and adjuvant therapy), for the different cutoff values, for disease status (metastatic and (neo)adjuvant disease), and for ethnicity (Caucasian and Ascian). A sensitivity analysis of the investigated outcomes was conducted by sequentially excluding each included study. Funnel plot asymmetry was used to detect publication bias, and Egger’s regression test was applied to assess the funnel plot for significant asymmetry [23]. The “trim and fill” method was used to test and adjust publication bias [24]. Statistical analyses were performed with STATA software (version 17.0; Stata Corporation, College Station, TX, USA). All *p* values were two-sided, with *p* < 0.05 considered statistically significant.

## 3. Results

### 3.1. Eligible Studies

The PRISMA flow diagram of the meta-analysis is shown in Figure 1. A total of 2258 records were identified from the systematic literature review, of which 990 duplicate records were removed. After title and abstract screening of the remaining records, 287 studies were selected for full-text review. After detailed evaluations, 247 studies were excluded due to outcomes, biomarkers, or other reasons. Eventually, 40 eligible studies were included in this meta-analysis [9,10,13,16,25,26,27,28,29,30,31,32,33,34,35,36,37,38,39,40,41,42,43,44,45,46,47,48,49,50,51,52,53,54,55,56,57,58,59,60]. The quality assessment of the included studies is summarized in Appendix A.

### 3.2. Study Characteristics

This meta-analysis included 12,229 participants from 40 eligible publications [9,10,13,16,25,26,27,28,29,30,31,32,33,34,35,36,37,38,39,40,41,42,43,44,45,46,47,48,49,50,51,52,53,54,55,56,57,58,59,60]. The characteristics of the studies are presented in Table 1. The studies were conducted between 2001 and 2021, 28 of which were prospective cohort studies, while the other 12 were retrospective cohort studies. In these studies, the proportion of patients with an elevated sHER2-ECD ranged from 4.40% to 78.99%. A majority of studies (n = 26) defined the cutoff value of an elevated sHER2 ECD level as 15 ng/mL.

There were 13 studies conducted in the (neo)adjuvant stage, 26 in metastatic settings, and the remaining 1 in both the (neo)adjuvant and metastatic phases. Regarding the treatments, 11 studies included patients treated with trastuzumab, 3 studies with patients receiving TKIs, 3 studies with endocrine therapy, and 9 studies with chemotherapy. As for ethnicity, 30 studies were Caucasian, and 10 were Asian.

### 3.3. Progression-Free Survival

A total of seventeen studies (n = 3662) reported the correlation between the sHER2 ECD level and PFS [10,13,16,25,27,28,31,32,33,39,44,45,48,49,53,54,59]. Between the studies, significant heterogeneity was seen with an I^2^ index of 91.2% (*p* < 0.001). The pooled analysis results indicated that an elevated sHER2 ECD level was significantly associated with shorter PFS (HR 1.74; 95% CI 1.40–2.17; *p* < 0.001; Figure 2) using the random-effects method. According to the sensitivity analysis, serial exclusion of studies was not significantly associated with the point estimates of pooled HRs for PFS (range, 1.68–1.80). Taking account of the excluded studies (Appendix A), the pooled HRs remained statistically significant. This suggested that the pooled results were not affected by any of the single studies included and that this meta-analysis had relatively stable results.

Subgroup analysis was performed according to different treatment modalities and cutoff values (Table 2). Interestingly, elevated levels of sHER2 ECD were significantly associated with poor clinical outcomes in patients treated with chemotherapy (pooled HR = 1.81, 95% CI 1.37–2.39, Appendix A), endocrine therapy (pooled HR = 1.91, 95% CI 1.57–2.32), and trastuzumab administration (pooled HR = 1.74, 95% CI 1.31–2.30). However, no such association was found in patients treated with TKIs (pooled HR = 1.44, 95% CI 0.85–2.43). As for different cutoff values, the prognostic significance could also be seen in the 15 ng/mL (*p* < 0.001), 15–20 ng/mL (*p* < 0.001), and 2500 U/mL (*p* = 0.011) group (Appendix A). Furthermore, the same result was also observed in Caucasians (pooled HR = 1.71, 95% CI 1.36–2.16) and Asians (pooled HR = 1.93, 95% CI 1.18–3.17) (Appendix A).

### 3.4. Disease-Free Survival

Thirteen studies with a total of 7599 patients offered adequate data for DFS analysis [9,13,29,37,38,43,46,50,51,55,56,57,60]. The pooled HR was 2.31 (95% CI 1.94–2.75, *p* < 0.001, Figure 3), suggesting that a high level of sHER2 ECD is highly correlated with poor DFS. There was no evidence of statistically significant heterogeneity between studies (I^2^ = 25.5%; *p* = 0.187). In the sensitivity analysis, the pooled HR estimates for DFS were not affected after excluding each study (Appendix A). The subgroup analysis revealed an association between an elevated sHER2 ECD level and a shorter DFS in all subsets (Table 2, Appendix A).

### 3.5. Overall Survival

A total of 20 studies, involving 5963 patients, provided applicable data for OS analysis [13,25,26,28,30,32,33,38,39,40,41,44,46,49,51,54,55,57,58,59]. As the heterogeneity among these studies showed statistical significance (I^2^ = 48.4%, *p* = 0.008), the random-effect model was employed to estimate the pooled HR (HR = 2.13, 95% CI 1.77–2.57, *p*  <  0.001, Figure 4). The results demonstrated that an elevated sHER2 ECD was significantly associated with unfavorable OS in breast cancer patients. In addition, after the sequential exclusion of each study from the pooled analysis, the conclusion was not changed (Appendix A). The results of each subgroup analysis that included different cutoff values, treatment modalities, ethnicities, and disease status were generally consistent with the entire patient cohort (Table 3, Appendix A).

### 3.6. Objective Response Rate

Funnel plots were generated to detect potential publication bias (Appendix A), which did not reveal remarkable asymmetry for DFS, OS, and ORR. In the PFS analysis, the funnel plot showed the asymmetric distribution of studies, and Egger’s test was significant (*p <* 0.001; Appendix A), suggesting that some publication bias might be present. Ten necessary studies were found missing by the “trim and fill” method. After filling in these ten with comprehensive analysis, the funnel plot regained symmetry (Appendix A). With the trim-and-fill approach, the imputed estimate HR and 95% CI was 1.65 (1.34 to 2.03), suggesting that publication bias should be considered, but it was not a major influencing factor for the intervention effect. In other words, the major outcomes were not affected by publication bias.

## 4. Discussion

Currently, a variety of monoclonal antibodies, TKIs, and antibody-drug conjugates have been approved for HER2-positive breast cancer, providing more options for these patients. However, there are few biomarkers for efficacy prediction and prognostic assessment in HER2-positive breast cancer. Some studies have shown that high-level sHER2 ECD was associated with significantly poor prognosis [47], but other studies could not confirm this finding [48]. To our knowledge, this is the first meta-analysis to explore the relationship between sHER2 ECD and different treatment regimens in breast cancer. Our results demonstrated that high levels of sHER2 ECD can be used as a biomarker for predicting a poor response to chemotherapy, endocrine therapy, and trastuzumab-targeted therapy, but they did not affect the efficacy of TKIs.

Our findings indicated that elevated sHER2 ECD was associated with poor clinical outcomes, which were consistent with the results of some previously published studies [19,26,60,61]. For example, in a large sample retrospective study of 2862 patients, elevated sHER2 ECD was an independent prognostic factor for worse distant-metastasis-free survival and breast-cancer-specific survival [57]. Moreover, in a study by Moreno et al [9], patients were randomly assigned to arms A (standard chemotherapy), B (standard chemotherapy with sequential trastuzumab), and C (standard chemotherapy with concurrent trastuzumab). The results showed that patients with baseline sHER2 levels ≥15 ng/mL had worse DFS than those with baseline sHER2 levels <15 ng/mL (arm A: HR, 1.81; *p* = 0.0014; arm B: HR, 2.08; *p* = 0.0015; arm C: HR, 1.96; *p* = 0.01) [9]. Additionally, for patients with metastatic breast cancer, an elevated ECD was significantly associated with decreased PFS and OS [16,25,32]. Of note, in our analysis, the baseline sHER2 ECD was not related to the ORR in the overall population. However, in the subgroup receiving endocrine therapy, patients with low sHER2 ECD levels had a higher ORR. A study by Colomer. et al. showed that in patients with metastatic breast cancer treated with letrozole, the ORR was lower in the group with elevated HER2 ECD levels (14% vs. 30%; *p* < 0.036) [25]. Similarly, in another study of patients treated with letrozole and tamoxifen, elevated serum HER-2/neu was a negative predictor for ORR and time to progression [27].

The value of 15 ng/mL was the cutoff for sHER2 ECD in most studies, but higher cutoff values were also found in some studies, especially in those for metastatic breast cancer [42,52]. In the current study, different cutoff values for sHER2 ECD were explored, with consistent results. It should be emphasized, however, that sHER2 ECD levels vary greatly in different stages of breast cancer. Among patients with early-stage breast cancer, only 4%–10% had sHER2 ECD levels greater than 15 ng/mL [31,57], while 20–80% of patients with metastatic breast cancer had an sHER2 ECD above 15 ng/mL [26,27,28,34]. Furthermore, the level of sHER2 ECD exhibited dynamic changes during treatment and disease progression. Metastatic breast cancer patients with decreased sHER2 ECD tend to have longer PFS [48,62]. In addition, sHER2 ECD levels were higher in patients with tissue HER2-positive, ER-negative, axillary lymph node, and distant metastases [50]. 

We performed a subgroup analysis to evaluate whether the effects of the sHER2 ECD differed for the treatment modalities. In the subgroup analysis of different treatment regiments, patients receiving trastuzumab with high sHER2 ECD had worse DFS, PFS, and OS levels than those with low sHER2 ECD levels. Similar results were obtained in the subgroup receiving chemotherapy. Interestingly, however, levels of sHER2 ECD were not associated with PFS and OS in TKI-treated patients. An elevated baseline sHER2 ECD implied resistance to trastuzumab but did not predict the response to lapatinib [45]. Lee et al reported that elevated sHER2 ECD predicted greater PFS benefit with lapatinib independent of sHER2 status [63]. The above evidence suggests that TKIs may overcome the drug resistance and disease progression caused by HER2 ECD shedding. A constitutively active truncated receptor, p95 HER2, is left by ECD shedding in the membrane, and it is 10-100-fold more oncogenic than the full-length receptor [4]. Cleavage of the ECD from HER2 loses the target that trastuzumab and pertuzumab bind to, while it significantly increases the tyrosine kinase activity of the truncated receptor and substantially enhances its transforming potential, which may be the mechanism by which the efficacy of TKIs is not affected. Therefore, the sHER2 ECD level may help clinicians select the appropriate anti-HER2 therapy for patients with HER2-positive breast cancer.

As a meta-analysis, there were limitations to our study. First, the funnel plot analysis of PFS showed an asymmetric distribution, implying the existence of publication bias. Second, the number of studies using TKIs as treatment regimens was relatively small. Prospective clinical studies with larger samples are needed for further validation. Despite these limitations, this systematic review and meta-analysis revealed that sHER2 ECD is an important prognostic factor for DFS, PFS, and OS in breast cancer patients and found that TKIs can overcome the adverse effects of elevated sHER2 ECD.

## 5. Conclusions

In summary, this systematic review and meta-analysis showed that elevated sHER2 ECD levels might be a prognostic indicator for reduced DFS, PFS, and OS. However, elevated sHER2 ECD levels did not affect the efficacy of TKIs. The baseline cutoff value of sHER2 ECD in different treatment stages of breast cancer needs to be further explored, and the combination of multi-node dynamic monitoring is more conducive to monitoring the treatment efficacy and predicting the prognosis of patients.

## Figures and Tables

**Figure 1 cancers-14-04551-f001:**
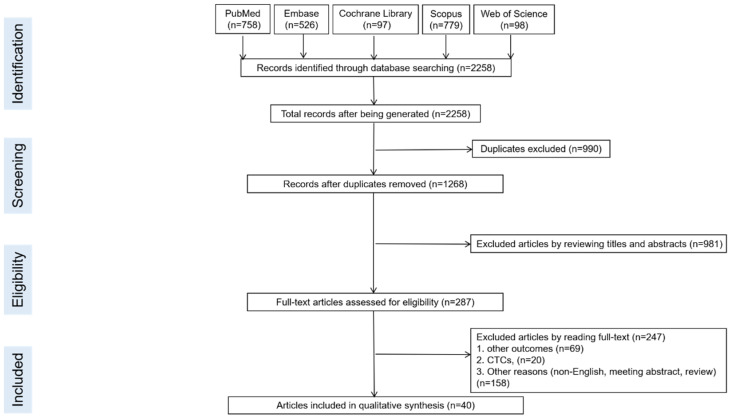
Flow diagram illustrating the selection of the included studies.

**Figure 2 cancers-14-04551-f002:**
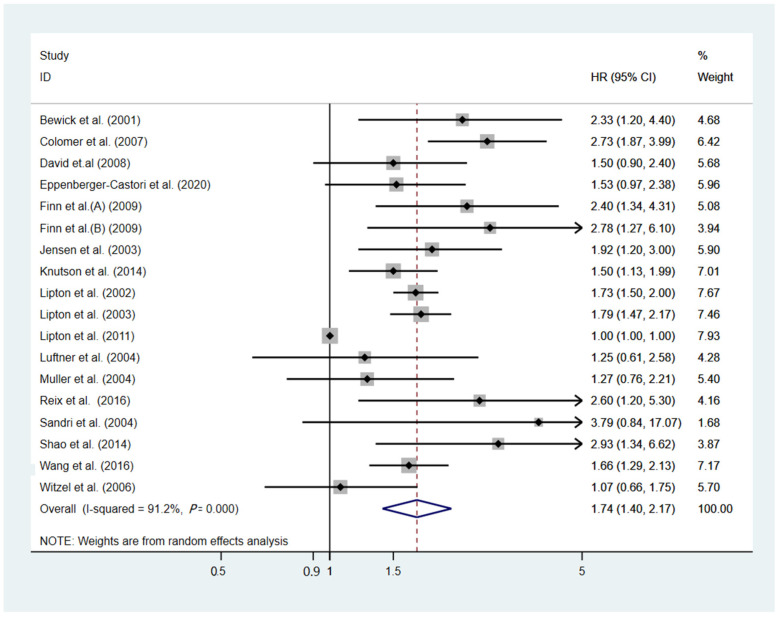
Forest plot of the HR for the PFS from the eligible studies [10,13,16,25,27,28,31,32,33,39,44,45,48,49,53,54,59].

**Figure 3 cancers-14-04551-f003:**
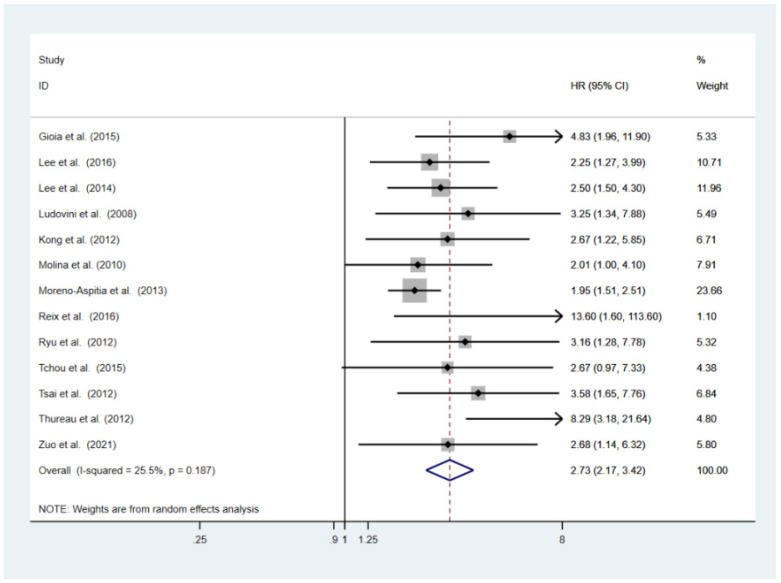
Forest plot of the HR for the DFS from the eligible studies [9,13,29,37,38,43,46,50,51,55,56,57,60].

**Figure 4 cancers-14-04551-f004:**
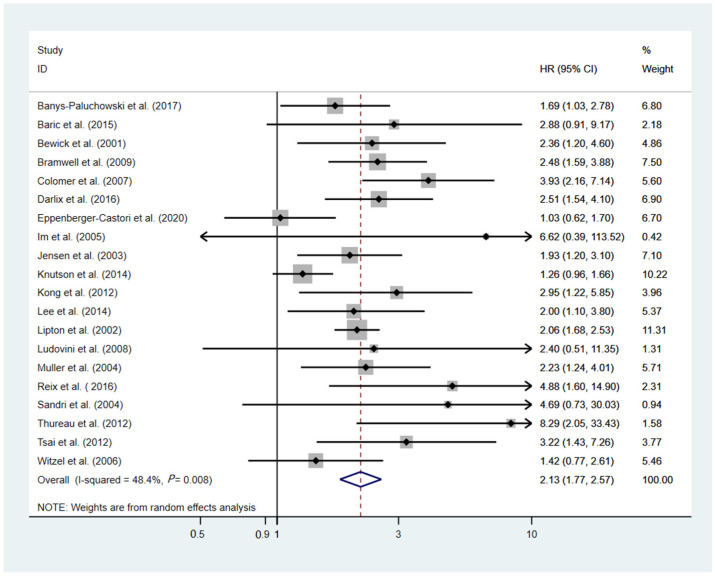
Forest plot of the HR for the OS from the eligible studies [13,25,26,28,30,32,33,38,39,40,41,44,46,49,51,54,55,57,58,59].

**Table 1 cancers-14-04551-t001:** Baseline characteristics of the studies included in the systematic review and meta-analysis.

Study	Year	Sample Size	Elevated (%)	Ethnicity	Study Design	Disease Status	Treatment	Outcomes	Cutoff of ECD	Exposure Assessment
Banys et al. [26]	2017	25	47.04	Caucasian	Prospective	Metastatic	NA	OS	15 ng/mL	ELISA
Baric et al. [30]	2015	79	44.30	Caucasian	Retrospective	Adjuvant	Adjuvant treatment	OS	15.86 ng/mL	ELISA
Bewick et al. [59]	2001	46	43.48	Caucasian	Retrospective	Metastatic	Chemotherapy	PFS, OS	2500 U/ml	ELISA
Bramwell et al. [58]	2009	158	21.52	Caucasian	Prospective	Metastatic	NA	OS	15 ng/mL	ELISA
Colomer et al. [25]	2007	226	18.58	Caucasian	Prospective	Metastatic	Endocrine therapy	PFS, ORR, OS	20 ng/mL	ELISA
Colomer et al. [52]	2004	42	28.57	Caucasian	Prospective	Metastatic	Chemotherapy	ORR	30 ng/mL	ELISA
Colomer et al. [42]	2006	48	29.17	Caucasian	Prospective	Metastatic	Chemotherapy	ORR	50 ng/mL	ELISA
David et.al. [45]	2008	198	25.00	Caucasian	Prospective	Metastatic	TKIs	PFS	82 ng/mL	ELISA
Darlix et al. [41]	2016	250	25.20	Caucasian	Retrospective	Metastatic	NA	OS	30 ng/ml	ELISA
Gioia et al. [60]	2015	241	12.03	Caucasian	Retrospective	Adjuvant	Chemotherapy + Trastuzumab	DFS	15 ng/mL	Chemiluminescence
Eppenberger et al. [32]	2020	131	67.18	Caucasian	Prospective	Metastatic	Chemotherapy + Trastuzumab	OS, PFS	15 ng/mL	Chemiluminescence
Esteva et al. [47]	2002	30	70.00	Caucasian	Prospective	Metastatic	Chemotherapy + Trastuzumab	ORR	14.9 ng/mL	ELISA
Finn et al. [48]	2009	579	16.58	Caucasian	Prospective	Metastatic	TKIs/Chemotherapy	ORR, PFS	16 ng/ml	ELISA
Fornier et al. [36]	2005	55	69.09	Caucasian	Retrospective	Metastatic	Chemotherapy + Trastuzumab	ORR	15 ng/mL	Chemiluminescence
Im et al. [40]	2005	27	14.81	Asian	Prospective	Metastatic	Chemotherapy	ORR, OS	15 ng/ml	ELISA
Jensen et al. [33]	2003	100	32.00	Caucasian	Prospective	Metastatic	Chemotherapy	OS, PFS	15 ng/ml	ELISA
Knutson et al. [54]	2014	54	55.56	Caucasian	Prospective	Metastatic	Chemotherapy + Trastuzumab	PFS, OS	15 ng/mL	ELISA
Kong et al. [46]	2012	252	15.08	Asian	Prospective	Adjuvant	Adjuvant treatment	DFS, OS	15 ng/mL	Chemiluminescence
Kontani et al. [35]	2013	19	63.16	Asian	Prospective	Metastatic	Chemotherapy + Trastuzumab	ORR	15.2 ng/mL	Chemiluminescence
Kostler et al. [34]	2004	55	72.73	Caucasian	Prospective	Metastatic	Chemotherapy + Trastuzumab	ORR	15 ng/mL	ELISA
Lee et al. [56]	2016	436	11.93	Asian	Retrospective	Adjuvant	Adjuvant treatment	DFS	15 ng/mL	Chemiluminescence
Lee et al. [57]	2014	2862	4.40	Asian	Retrospective	Adjuvant	Adjuvant treatment	DFS, OS	15.2 ng/mL	Chemiluminescence
Lipton et al. [44]	2002	719	30.46	Caucasian	Retrospective	Metastatic	Endocrine therapy	ORR, OS, PFS	15 ng/mL	ELISA
Lipton et al. [27]	2003	562	29.18	Caucasian	Prospective	Metastatic	Endocrine therapy	ORR, PFS	15 ng/mL	ELISA
Lipton et al. [10]	2011	138	78.99	Caucasian	Prospective	Metastatic	TKIs	PFS	15 ng/mL	ELISA
Ludovini et al. [38]	2008	256	8.98	Caucasian	Prospective	Adjuvant	Adjuvant treatment	OS, DFS	15 ng/mL	Chemiluminescence/ELISA
Luftner et al. [53]	2004	35	62.86	Caucasian	Prospective	Metastatic	Chemotherapy	ORR, PFS	15 ng/mL	ELISA
Molina et al. [37]	2010	275	14.91	Caucasian	Prospective	Adjuvant	Adjuvant treatment	DFS	15 ng/mL	ELISA
Moreno-Aspitia et al. [9]	2013	2318	37.41	Caucasian	Retrospective	Adjuvant	Chemotherapy ± Trastuzumab	DFS	15 ng/mL	Chemiluminescence
Muller et al. [49]	2004	103	35.92	Caucasian	Retrospective	Metastatic	Chemotherapy	ORR, PFS, OS	15 ng/mL	ELISA
Reix et al. [13]	2016	334	15.27	Caucasian	Prospective	(Neo)adjuvant/Metastatic	Chemotherapy + Trastuzumab	DFS, PFS, OS	15 ng/mL	ELISA
Ryu et al. [50]	2012	200	7.00	Asian	Prospective	Adjuvant	Adjuvant treatment	DFS	15 ng/mL	Chemiluminescence
Sandri et al. [28]	2004	39	10.26	Caucasian	Prospective	Metastatic	Chemotherapy	PFS, OS	15 ng/mL	Chemiluminescence
Shao et al. [31]	2014	62	41.94	Asian	Prospective	Metastatic	Chemotherapy + Trastuzumab	PFS, ORR	15 ng/mL	Chemiluminescence
Tchou et al. [29]	2015	118	22.88	Caucasian	Prospective	Adjuvant	Adjuvant treatment	DFS	7 ng/mL	ELISA
Thureau et al. [51]	2012	65	10.77	Caucasian	Retrospective	Adjuvant	Chemotherapy + Trastuzumab	DFS, OS	15 ng/mL	ELISA
Tsai et al. [55]	2012	185	12.43	Asian	Prospective	Adjuvant	Adjuvant treatment	DFS, OS	8.9 ng/mL	ELISA
Wang et al. [16]	2016	546	43.22	Asian	Prospective	Metastatic	NA	PFS	15 ng/mL	Chemiluminescence
Witzel et al. [39]	2006	76	39.47	Caucasian	Prospective	Metastatic	NA	PFS, OS	NA	NA
Zuo et al. [43]	2021	309	53.07	Asian	Retrospective	Neoadjuvant	Chemotherapy + Trastuzumab	DFS	15 ng/mL	Chemiluminescence

Abbreviations: ORR, overall response rate; DFS, disease-free survival; OS, overall survival; PFS, progression-free survival; NA, not available; ELISA, enzyme-linked immunosorbent assay.

**Table 2 cancers-14-04551-t002:** Subgroup analysis of the pooled HR for the PFS and DFS.

Categories	PFS						DFS					
	No. of Studies	No. of Patients	Pooled HR (95% CI)		Heterogeneity		No. of Studies	No. of Patients	Pooled HR (95% CI)		Heterogeneity	
			Random	*p*-Value	I² (%)	*p*-Value			Random	*p*-Value	I² (%)	*p*-Value
**All patients**	17	3662	1.74 (1.40–1.72)	<0.001	91.2	<0.001	13	7599	2.31 (1.94–2.75)	<0.001	25.5	0.187
**Treatment**												
Chemotherapy	6	611	1.81 (1.37–2.39)	<0.001	1.8	0.366	1	795	1.81 (1.26–2.61)		NA	NA
Endocrine therapy	3	1507	1.91 (1.57–2.32)	<0.001	59.2	0.086						
TKIs	3	627	1.44 (0.85–2.43)	0.17	82.2	0.004						
Trastuzumab + Chemotherapy	4	295	1.74 (1.31–2.30)	<0.001	24.3	0.265	5	2318	3.25 (1.98–5.31)		57.4	0.038
Adjuvant therapy							8	4332	2.60 (2/02–3.36)		0	0.967
**Cutoff value**												
15 ng/mL	12	2537	1.64 (1.27–2.11)	<0.001	92.2	<0.001	12	7260	2.68 (2.11–3.41)	<0.001	27.3	0.176
5–10 ng/mL							1	185	3.58 (1.65–7.76)	0.001	-	-
15–30 ng/mL	3	805	2.65 (1.97–3.56)	<0.001	0	0.928						
82 ng/mL	1	198	1.50 (0.92–2.45)	0.105	-	-						
2500 U/mL	1	46	2.33 (1.22–4.46)	0.011	-	-						
**Disease status**												
Metastatic	18	3662	1.74 (1.40–2.17)	<0.001	91.2	<0.001						
(Neo)adjuvant							13	7445	2.73 (2.17–3.42)	<0.001	25.5	0.187
**Ethnicity**												
Caucasian	16	3054	1.71 (1.36–2.16)	<0.001	91.2	<0.001	7	3317	2.59 (1.88–3.57)	<0.001	29.3	0.205
Asian	2	608	1.93 (1.18–3.17)	0.009	43.8	0.182	6	4128	2.95 (2.13–4.07)	<0.001	13.8	0.326

TKI, tyrosine kinase inhibitor; DFS, disease-free survival; PFS, progression-free survival; HR, hazard ratio; CI, confidence interval.

**Table 3 cancers-14-04551-t003:** Subgroup analysis of the pooled HR/OR for the OS and ORR.

Categories	OS						ORR					
	No. of Studies	No. of Patients	Pooled HR (95% CI)		Heterogeneity		No. of Studies	No. of Patients	Pooled or (95% CI)		Heterogeneity	
			Random	*p*-Value	I^2^ (%)	*p*-Value			Random	*p*-Value	I^2^ (%)	*p*-Value
**All patients**	**20**	5963	2.13 (1.77–2.57)	<0.001	48.4%	0.008	14	2274	0.80 (0.49–1.31)	0.381	73	< 0.001
**Treatment**												
Chemotherapy	5	315	2.19 (1.60–3.01)	< 0.001	0	0.817	7	605	0.69 (0.38–1.26)	0.225	43.6	0.1
Endocrine therapy	2	945	2.67 (1.44–4.97)	0.002	75.1	0.045	3	1507	0.37 (0.28–0.49)	< 0.001	0	0.988
TKIs							1	291	1.03 (0.50–2.13)	0.931	NA	NA
Trastuzumab + Chemotherapy	4	584	2.00 (1.02–1.95)	0.045	76.8	0.005	4	159	5.50 (1.15–26.21)	0.033	0.035	65
Adjuvant therapy	5	3382	2.56 (1.75–3.74)	< 0.001	0	0.896						
**Cutoff value**												
15 ng/mL	14	5101	1.98 (1.59–2.48)	< 0.001	51.1	0.014	10	1667	1.02 (0.53–1.97)	0.985	76.4	< 0.001
5–10 ng/mL	1	185	3.22 (1.43–7.26)	0.005	NA	NA						
15–30 ng/mL	3	555	2.99 (2.09–4.29)	< 0.001	0	0.523	3	847	0.46 (0.16–1.29)	0.138	73.7	0.022
50 ng/mL							1	48	1.06 (0.30–3.71)	0.924	NA	NA
2500 U/ml	1	46	2.36 (1.21–4.62)	0.012	NA	NA						
**Disease status**												
Metastatic	13	2182	1.94 (1.58–2.38)	< 0.001	54.8	0.009	13	2562	0.80 (0.49–1.31)	0.381	73	< 0.001
(Neo)adjuvant	6	3447	2.78(1.92–4.01)	< 0.001	0	0.605						
Metastatic and (Neo) adjuvant	1	334	4.88 (1.6–14.89)	0.005	NA	NA						
**Ethnicity**												
Caucasian	16	2889	2.07 (1.68–2.55)	< 0.001	55.2	0.004	11	2166	0.74 (0.46–1.20)	0.220	71.0	< 0.001
Asian	4	3074	2.58 (1.71–3.90)	< 0.001	0	0.689	3	108	2.63 (0.11–63.22)	0.522	85.2	0.001

TKI, tyrosine kinase inhibitor; ORR, overall response rate; OS, overall survival; HR, hazard ratio; OR, odd ratio; CI, confidence interval.

## Data Availability

The data presented in this study are available on request from the corresponding author.

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
