# Peer review of "Prognostic Value of the Serum HER2 Extracellular Domain Level in Breast Cancer: A Systematic Review and Meta-Analysis"

_cancers, 2022, doi:10.3390/cancers14194551_

Round 1

Reviewer 1 Report

In the present manuscript entitled “Prognostic value of serum HER2 extracellular domain level in breast cancer: a systematic review and meta-analysis” the authors have performed a meta-analysis to investigate the correlation between serum HER2 extracellular domain and breast cancer. This is a significant study with sufficient methodology as the authors performed a comprehensive literature search and conducted the meta-analysis based on PRISMA statement.

The manuscript is well constructed, well written and well documented and can be potentially accepted. There are a few minor issues that the authors should address, in order to ameliorate their manuscript and publish their results.

1. With respect to the data extraction and quality assessment of the included studies, the authors clarified that 3 researchers conducted the study selection and data extraction. However, only three electronic databases (PubMed, Embase, Cochrane Library) were systematically searched for eligible studies. The small number of acquired studies could be responsible for the existence of publication bias reported in this meta-analysis. Did the researchers search on other databases such as Web of Science or Scopus?

2. The authors should be more specific concerning the inclusion and exclusion criteria of their meta-analysis. For example, other meta-analyses were included? Data based on animal samples were excluded?

3. A table displaying the most important findings of the meta-analysis is suggested.

4. Please include citations for all the studies used in this meta-analysis.

Author Response

  1. With respect to the data extraction and quality assessment of the included studies, the authors clarified that 3 researchers conducted the study selection and data extraction. However, only three electronic databases (PubMed, Embase, Cochrane Library) were systematically searched for eligible studies. The small number of acquired studies could be responsible for the existence of publication bias reported in this meta-analysis. Did the researchers search on other databases such as Web of Science or Scopus?

Response: Thanks for the insightful comment.

The suggestion has been taken seriously. In the previous analysis, we searched PubMed, Embase, Cochrane Library databases for this systematic review. To ensure inclusion of relevant literature, an additional database search was performed using the Web of Science and Scopus database (Sept 11th 2022) to confirm that the initial literature coverage was comprehensive. The detailed search strategies were available in Supplementary Materials Table S1.

A total of 2258 records were identified from the systematic literature review, of which 990 duplicate records were removed. After title and abstract screening and detailed evaluations, 40 eligible studies were included in this meta-analysis. No additional eligible articles were identified in the Web of Science and Scopus database search. Please see Methods and Results Section on Page 2 Line 96 and Page 4 Line 167-170 and revised Figure 1.

Thanks for your valuable suggestion again.

  1. The authors should be more specific concerning the inclusion and exclusion criteria of their meta-analysis. For example, other meta-analyses were included? Data based on animal samples were excluded?

Response: We appreciate this constructive comment.

We have elaborated the inclusion and exclusion criteria based on Population, Intervention, Comparison, Outcomes and Study designs (PICOS) framework in the revised manuscript. Additionally, meta-analyses and non-human research such as animal or experimental studies were excluded. We have addressed this part in the Methods Section (Page 3 Line 103-114).

Thanks for your prudent advice again.

  1. A table displaying the most important findings of the meta-analysis is suggested.

Response: We really appreciated your valuable advice. For clarity, the main findings for the overall and the subgroup populations were displayed in the revised table 2 and table 3.

  1. Please include citations for all the studies used in this meta-analysis.

Response: Thanks for your careful work.

We really apologize for this oversight. In the revised manuscript, we have inserted the citations for all included studies and revised these mistakes throughout. Please see Result Section on Page 4 Line 172, 177-178, Page 6 Line 259, Page7 Line 291, Page 8 Line 303,315 and revised Figure 1.

Reviewer 2 Report

The authors performed meta-analysis of the predictive value of serum HER2 extracellular domain as a possible biomarker for the prognosis of breast cancer clinical course.

They obtained statistically significant results that the high level of sHER2 EBD highly correlated with poor DFS, was significantly associated with unfavorable OS in breast cancer, was significantly associated with lower ORR and poor prognosis after treatment with Herceptin.

They found also that levels of sHER2 EBD can be used as a biomarker for predicting poor response to chemotherapy, endocrine therapy and trastuzumab – targeted therapy but did not affect the efficacy of tyrosine kinase inhibitors therapy.

The authors gave also the explanation of the results they described at the molecular level.

The meta-analysis was conducted on 12229 participants. The authors should also add the information about the ethnic groups included in the study.

Author Response

Reviewer 2

The authors performed meta-analysis of the predictive value of serum HER2 extracellular domain as a possible biomarker for the prognosis of breast cancer clinical course.

They obtained statistically significant results that the high level of sHER2 EBD highly correlated with poor DFS, was significantly associated with unfavorable OS in breast cancer, was significantly associated with lower ORR and poor prognosis after treatment with Herceptin.

They found also that levels of sHER2 EBD can be used as a biomarker for predicting poor response to chemotherapy, endocrine therapy and trastuzumab – targeted therapy but did not affect the efficacy of tyrosine kinase inhibitors therapy.

The authors gave also the explanation of the results they described at the molecular level.

The meta-analysis was conducted on 12229 participants. The authors should also add the information about the ethnic groups included in the study.

Response: We deeply appreciate your positive evaluation of our work.

The suggestion has been taken seriously. We have added the race information in revised Table 1 and have addressed this part in the Methods and Results section of the revised manuscript (Page 3 Line 141 and Page 4 Line 186-187). In addition, we also performed subgroup analysis for different ethnic groups. Likewise, the results of different ethnicities were similar to those of the overall patients. Please see Result Section on Page 6 Line 276-277 and Page 8 Line 310, 322; revised Table 2-3 and Supplementary Materials Figure S2-5.

Thanks for your valuable suggestion again.

We would like to thank you for your comments that have helped us to improve our manuscript. We hope that the revised manuscript will be satisfactory for publication in Cancers.
